# Detecting Snowfall Events over the Arctic Using Optical and Microwave Satellite Measurements

Emmihenna Jääskeläinen[1], Kerttu Kouki[1], and Aku Riihelä[1]

[1]Finnish Meteorological Institute, Erik Palménin aukio 1, Helsinki

**Correspondence:** Emmihenna Jääskeläinen (emmihenna.jaaskelainen@fmi.fi)

**Abstract.** The precipitation over the Arctic region is a difficult quantity to determine with high accuracy, as the in situ observation network is sparse, and current climate models, atmospheric reanalyses and direct satellite-based precipitation observations suffer from diverse difficulties that hinder the correct assessment of precipitation. We undertake a proof-of-concept investigation into how accurately optical satellite observations, namely Sentinel-2 surface reflectance-based grain size-connected specific surface area of snow (SSA), and microwave-based snow water equivalent (SWE) estimates can detect snowfalls over the Arctic. In addition to the satellite data, we also include ERA5-Land SWE data to support the analysis. Here, we chose a limited area (a circle of 100 km radius around Luosto radar located in Northern Finland) and a short time period (covering March 2018) to test these data sources and their usability to this precipitation assessment problem. We classified differences between observations independently for SSA and SWE and compared the results to the radar-based snowfall information. These initial results are promising. Situations with snowfalls are classified with high recalls, 64% for the satellite-based SWE, 77% for ERA5-Land-based SWE and around 90% for SSA when compared to radar-based data. Cases without snowfalls are more difficult to classify correctly by using satellite-based data. The recall values are 34% for satellite-based SWE and varying from almost 60% to over 70% for SSA. SWE from ERA5-Land has the highest recall value for cases without snowfall, 80%. These results indicate, that optical and microwave-based satellite observations can be used to detect snowfall events over the Arctic.

## 1 Introduction

Precipitation in all its forms drives the hydrological cycle over land and it is also responsible for the mass balance of glaciers and ice sheets. Precipitation in the form of snow creates and grows the seasonal snowpack over the high latitudes of the Northern Hemisphere. The future of this seasonal snow depends largely on the Arctic temperature regime and trends; climate models of both the fifth and sixth phases of the Coupled Model Intercomparison Project (CMIP5 and CMIP6, respectively) are projecting an increase in rainfall and consequently a decrease in snowfall over the Arctic with increasing warming (Bintanja and Selten (2014), Vihma et al. (2016), Bintanja and Andry (2017), McCrystall et al. (2021)). However, climate models in general struggle to match observed warming over the Arctic during the past decades (Rantanen et al. (2022)).

Atmospheric reanalyses provide continuous coverage of precipitation, supported by broad assimilation of observation data of the atmospheric state. However, assessments of modern reanalyses over the Arctic Ocean have found a wide range of

portrayed frequency, intensity, and annual or seasonal total precipitation (Boisvert et al. (2018), Barrett et al. (2020)). Given this variation and the linkages between changes in the state of the Arctic sea ice and the high-latitude hydrological cycle (Screen and Simmonds (2013), Merkouriadi et al. (2017), Sato and Inoue (2018), Webster et al. (2018)), it is natural that reanalysis- and model-based precipitation estimates over the Arctic land masses will also exhibit substantial variability (Krasting et al. (2013), Kouki et al. (2022)).

Direct satellite-based observations of high-latitude precipitation now exist, measured by radars on board the Global Precipitation Mission (GPM) and CloudSat satellite missions. While recent progress in quantification of the Arctic snowfall from these sources has been made (Edel et al. (2020), Skofronick-Jackson et al. (2019)), the limited swath coverage of spaceborne radars, combined with the small amount of available observing platforms, is the main reason for limited spatiotemporal coverage.

Tracking Arctic snowfall events on a wider spatiotemporal scale is thus possible either by correctly modeling the atmospheric conditions that generate snowfall events or by detecting falling snow in the atmosphere using weather radar observations. Yet, fallen snow also modifies the observable surface properties such as surface reflectivity, albedo, or snow water equivalent (SWE). Detection of autumn's first snowfall over the seasonal snow zone is trivial because of the stark albedo difference between snowy and non-snowy land surfaces. A more challenging task is to detect fresh snow atop older snow. One solution is to use grain size information. We expect that fresh snow deposited in snowfall events is typically smaller-grained and thus more reflective than any existing aged snow cover surface (Legagneux et al. (2002), Flanner and Zender (2006), Taillandier et al. (2007)). Therefore, the possibility exists to detect snowfall events *a posteriori* by investigating changes in optical satellite imagery related to snow reflectivity and grain size properties. Another possible solution for detecting snowfall events is to use microwave-based SWE, which is the amount of water contained in the snowpack (in units of $kg/m^2$) or equivalently, the height of the water layer (in units of mm) that would result from melting the whole snowpack instantaneously (Fierz et al. (2009)). Therefore, when snow falls, it is expected that SWE will increase, provided that no melting or sublimation occurs.

The aim of this study is to investigate if the detection of snowfall events (in terms of occurrence, not intensity) is possible from satellite observations indirectly, using two methods: 1) From high-resolution satellite imagery covering the visible and near-infrared wavebands using the 'footprint' they leave on the surface properties of snow, and 2) from abrupt increases in daily SWE, based on microwave emissions from the snow cover. However, the challenges in this investigation are numerous. Optical imagery is only available under clear skies, potentially extending the pre/post-snowfall sampling period and diminishing the detectable change. Lack of sufficient sunlight during late autumn and winter over high latitudes also effectively limits our investigation to spring period snowfalls. The microwave satellites, in turn, only provide data at a coarse resolution, which can complicate the analysis. The investigation also requires a robust reference dataset for the occurrence of snowfall to be feasible; for this purpose, we employ spatiotemporally well-resolved ground-based weather radar measurements from the radar network of the Finnish Meteorological Institute (FMI) over Finnish Lapland.

This paper is structured as follows. We begin by describing the area of investigation and the chosen satellite imagery, as well as the weather radar data serving as a reference, and their pre-processing methods (Sections 2 and 3). Supporting data from atmospheric reanalysis and other auxiliary sources are also described. Obtained results are then presented in Section 4, followed by a summarizing discussion on the strengths and weaknesses of the investigated approach in Section 5.

## 2  Data

The study area of this work is located in Northern Finland, a circle of 100 km radius around the weather radar placed on Luosto fell (centered at 67.1386°N, 26.9008°E; Figure 1, panel a), and the chosen time period is March 2018. This particular area and this particular time period are chosen, because they fulfill at the same time both of the two necessary conditions: 1) solar zenith angle (SZA) is high enough in early spring to enable optical satellite-based observations, and 2) the temperature remains below -5°C most of the time period (only just at the end of the month the temperatures rise above -5°C), meaning that the precipitation falls as snow, and we do not have to take into account the metamorphosis of snow or snowmelt (e.g. Pirazzini (2004), Pirazzini et al. (2006), Kouki et al. (2019)).

We use both microwave and optical satellite data together with radar observations. Microwave-based SWE estimates and optical-based specific surface area (SSA) estimates (which are calculated from the surface reflectance data) are chosen as satellite-based data because they both are affected by snowfall. The reference snowfall data in our study are based on snowfall information from weather radar data. In addition to the satellite and radar data, we also include ERA5-Land SWE data to support the analysis.

### 2.1  Satellite data

Atmospheric corrected surface reflectance values are retrieved from the observations of MultiSpectral Instruments (MSI), onboard Sentinel-2 (S2) A and B satellites (ESA (2021)). These level-2A (L2A) data are available in 12 different wavelength bands, covering the visible, near-infrared (NIR), and short-wave infrared (SWIR) wavelength ranges. Compared to many other satellite-based data, the L2A data are provided in three very fine spatial resolutions: 10 m, 20 m, and 60 m, the resolution depending on the wavelength band. The L2A data are divided into predefined tiles, each of them consisting of ortho-images in UTM/WGS84 projection covering $110 \times 110$ km$^2$ areas. Each tile overlaps with the neighboring ones about 10 km.

In this study, we use data from band 9 (central wavelength 945 nm) with a spatial resolution of 60 m. The uncertainty for this band is not provided directly, but based on uncertainty for other wavelength channels, we assume that the uncertainty for band 9 is around 0.03 (Clerc and Team (2022)).

The satellite-based SWE dataset used in this study is the ESA CCI-Snow "SnowCCI" (European Space Agency Climate Change Initiative, Snow) version 2 data (Luojus et al. (2022)). The algorithm combines satellite-based microwave brightness temperature data with in situ snow depth measurements. To estimate SWE, the algorithm uses the difference in microwave brightness temperature between two frequencies (37 and 19 GHz). The different frequencies penetrate through the snowpack differently and, therefore, a large difference between the high-frequency and low-frequency signal indicates a larger snow volume. The algorithm combines satellite data with in situ measurements which notably improves the SWE estimates relative to a satellite-only retrieval (Pulliainen (2006)). Version 2 uses dynamic snow density, which improves the seasonal evolution of SWE (Mortimer et al. (2020)), making it well-suited for this study. The SnowCCI v2 is mapped to a 0.1° resolution and it is a daily SWE product, allowing us to detect daily changes in SWE.

## 2.2 Radar data

Finnish Meteorological Institute maintains a ground-based radar network, which consists of 11 dual-polarization C-band Doppler radars, spatially covering almost the whole area of Finland (FMI (2023)). Dual-polarization radars send out horizontally and vertically polarized electromagnetic waves, which are scattered when encountering particles and objects in the atmosphere. The radars receive these backscattered signals, which are then modified to different quantities using dedicated algorithms (Kumjian (2013)). One of the advantages of dual-polarized radars is that they are useful in identifying different precipitation types during winter. For example, snow particles have a uniform shape and size, which can be seen as a high (above 0.97) correlation coefficient value ($\rho_{hv}$) between polarizations (Kumjian (2013)).

Radar reflectivity $dBZ$ and correlation coefficient $\rho_{hv}$, observed every 10 minutes, were chosen for this study. Parameter $dBZ$, a decibel quantity derived from the radar reflectivity factor $Z$, is a precipitation intensity measurement. Higher $dBZ$ signifies stronger precipitation, and it can be used to calculate for example rainrate and snowrate (i.e. the amount of precipitation measured as mm/hr). In this study, the radar data are from Luosto radar which is located in Northern Finland.

## 2.3 ERA5 and ERA5-Land reanalysis

The fifth generation of European Centre for Medium-Range Weather Forecasts (ECMWF) atmospheric reanalysis of the global climate (ERA5), produced by the Copernicus Climate Change Service, covers the period spanning from 1940 up to the present day (Hersbach et al. (2020)). The ocean, land, and atmospheric variables are provided in 31 km spatial resolution and in three different time resolutions (hourly, daily, and monthly). In addition, the atmospheric variables are provided in multiple pressure levels, starting from the surface and going up to 80 km in the atmosphere. In this work, we use hourly data of eastward wind component $u$ (m/s), northward wind component $v$ (m/s), and geopotential $\phi(z)$ (m$^2$/s$^2$) to determine the wind drift trajectories of the snowfall. The wind components are the horizontal speeds of air moving either towards the east or north if the values are positive, or towards the west or south if the values are negative. Geopotential is the gravitational potential energy of a unit mass, and it can be used to calculate geopotential height (Holton (2004)):

$$H = \phi(z)/g_0, \tag{1}$$

where $z$ is a geometric height, and the global average of gravity at mean sea level $g_0$ is a constant value of 9.80665 m/s$^2$. Near the surface, the geopotential height can be considered to be numerically equal to the geometric height (Holton (2004)). The minimum detection height of the Luosto radar increases with the distance from the radar, reaching the maximum height of around 1.1 km above the level of the radar at the edge of the 100 km study area. Therefore, we can use the geopotential heights directly as the heights of the wind component layers, i.e. $H = z$.

In addition to the satellite-based SWE, we also include ERA5-Land (ERA5L) SWE data to the analysis. ERA5L is the land component of ERA5 (Muñoz-Sabater et al. (2021)). The spatial resolution of ERA5L is 9 km and, contrary to ERA5, ERA5L is run without data assimilation and coupling the atmospheric module. In this study, we use hourly SWE values from which we calculate daily means.

## 2.4 Auxiliary data

Auxiliary data on forest and terrain are required for the analysis and obtained from external sources as follows. The operational Finnish Multisource National Forest Inventory (MS-NFI) describes Finnish forests at a national scale from a variety of data sources. The first edition was created in 1990, and it has since been updated frequently. The national forest inventory provides multiple parameters, and the one we are interested in is the total canopy cover estimates of the trees [%]. In this work, we used the version based on Sentinel-2A MSI satellite images (bands 2, 3, and 4) from the year 2017 and an improved k-NN method (ik-NN). For more details, see for example Tomppo et al. (2013).

Our terrain is described with a Digital Elevation map (DEM) from the National Land Survey of Finland. These data are based on airborne laser scanning, and we use a version provided in 10 m spatial resolution. The total canopy cover and DEM of the study area are shown in Figure 1, panels b and c, respectively.

## 3 Preprosessing

All data are first reprojected to UTM zone 35N projection. Then all S2-related data are resampled to spatial resolution of 1 km × 1 km, and all SWE-related data are resampled to spatial resolution of 5 km × 5 km.

### 3.1 Snowrate calculation

Radar reflectivity $dBZ$ is processed to liquid equivalent snowrate using the so-called Z-R relationship (Marshall and Palmer (1948)):

$$R = \left(\frac{Z}{A}\right)^{-b}, \tag{2}$$

where $R$ is a snowrate (mm/h), $Z = 10^{\frac{dBZ}{10}}$, and coefficients $A$ and $b$ are determined empirically for snowfall. In our case, for Finland, the coefficients are $A = 115$ and $b = 1.35$. Due to the chosen 10-minute intervals and the unit of snowrate being mm/h, we divide the snowrate value by six to acquire the amount of snowfall for each individual time step.

After acquiring the snowrate values, they are screened using the condition $\rho_{hv} \geq 0.98$ to ensure that only snow observations are used. If $\rho_{hv} < 0.98$, we assume that there is no snowfall (the $R$ value is set to zero). A small spatial interpolation is performed to instantaneous snowrate values to remove the small gaps, from one to two missing data pixels due to the ground clutter, to provide more spatially continuous snowrate values. For persistent ground clutter areas (i.e. areas which have ground clutter almost always) a mask is created, and it is used to discard those pixels from the analysis.

### 3.2 Wind adjusted snowrate data

Snowfall can drift significantly due to the wind after it has been detected by the radar and before it hits the ground. Therefore, snowrate data need to be wind adjusted. For that, we use the minimum height of radar observation values, adjusted DEM, and $u$ and $v$ wind components and geopotential height data from the ERA5 reanalysis.

The minimum observation height data for the Luosto radar are provided as discrete values. The values are dependent on the distance of the radar, and therefore a simple second-degree polynomial fit is performed to obtain a function ($y = -0.04 + 5.42 \cdot 10^{-3} x + 5.77 \cdot 10^{-5} x^2$, where $x$ is the distance from the radar in km) to be used to calculate minimum observation height values for each pixel for the 100 km area around the Luosto radar. All predicted values below zero are assumed to be equal to zero. After calculating the radar minimum height values, the radar tower height (19 m) and DEM value for the location of the radar tower (240 m) are added to the values to lower the minimum observation height to sea level.

For the wind adjustment, the DEM data and ERA5-based data are modified. DEM is slightly adjusted by adding 10 m to every pixel which has a total canopy cover estimate above 10%. This is to add an assumption of around 10 m tall trees to all forested pixels to help determine the lowest geopotential height to be used (snowfall cannot be moved below ground level). The tree height of around 10 m is based on the environment information for Sodankylä forest (FMI (2022)). And because wind components, $u$ and $v$, and geopotential height $z$ data layers are provided as hourly data, they are interpolated temporally to correspond to 10-minute intervals of radar data.

The wind adjustment is performed separately for every 10-minute interval observation. We follow the method described in Lauri (2010). We do the wind adjustment from the ground up, meaning that we start from a blank matrix and fill that for snowrate values based on the wind drift. That way we only have one value in each pixel and we avoid discontinuities in the wind-adjusted data. More details of the algorithm are in Appendix A.

## 3.3 Preprocessing S2 data

S2 MSI data are from Copernicus Collection 1 (ESA (2021)), preprocessed with Google Earth Engine (Gorelick et al. (2017)), and provided as individual tiles. The multi-size mosaic tool by SeNtinel's Application Platform (SNAP; https://step.esa.int/main/) is used to process the overlapping 10 km areas of the identically time-stamped tiles by combining overlaying pixels. Data are divided back into tiles during the reprojecting and resampling phase (tile bounds are based on original tiles).

### 3.3.1 Cloud shadow removal

Clouds and cloud shadows from S2 data are removed by using the Copernicus Sentinel-2 Cloud Probability dataset, based on the gradient boosting-based sentinel-2-cloud-detector algorithm (Zupanc (2017)). The identification and removal of clouds is applied as part of the preprocessing in Google Earth Engine. As always, cloud detection over bright snow with probabilistic methods implies a trade-off between sampling and robustness, as bright surfaces seldom provide near-zero cloud probabilities. Here, a 30% cloud probability was chosen as the cutoff as a compromise between residual cloud contamination and sufficient sampling across our study domain. Also, to account for most cloud shadows in the imagery, we projected 9 km long cloud shadows and discarded imagery in the affected pixels. The effort is of course approximative as robust cloud top height data is unavailable from Sentinel-2 imagery alone. To limit these residual effects to the classification results, we further remove all those pixels that have at least one missing pixel due to the cloud contamination in neighboring pixels. This process is iterated only twice, due to the trade-off between losing some of the good quality data and not discarding enough cloud-contaminated pixels.

### 3.3.2 Forest correction

Luosto radar site and the study area around it are located in the boreal forest zone and have mostly evergreen needleleaf trees. The forest canopy complicates the detection of snow property changes in two ways. First, the boreal needleleaf canopy is dark, with typical (winter) albedo between 0.1 and 0.15 (Betts and Ball (1997)). For the near-nadir S2 imagery, this means that the snow-free canopy darkens the scene by its coverage fraction and complicates the detection of reflectivity changes in the under-canopy snow. Falling snow that is intercepted by the canopy may also, under the right conditions, remain on the branches for extended periods of time. This significantly brightens the canopy-covered area and thus the reflectivity of the scene.

In order to take both effects into account, we calculate a linear regression between independent forest canopy cover estimates and observed S2 reflectivities for each image. Then, the (snow) surface reflectance corrected for canopy darkening is calculated by subtracting forest density values multiplied by the image-mean slope term from the original snowrate (SR) values, that is:

$$\text{SR} = \beta_0 + \beta_1 \cdot \text{CC}$$

$$\text{SR}_{\text{corr}} = \text{SR} - \beta_1 \cdot \text{CC} \tag{3}$$

where CC is forest canopy cover, and $\beta_0$ and $\beta_1$ are linear regression coefficients. An example of forest correction is shown in Figure 2. To account for the possible snow interception on the canopy, the correction is applied for snowrate only if the corrected value remains below 1.0, as the snowy canopy would be overcorrected by this method which assumes the canopy to be snow-free.

### 3.3.3 SSA calculation

Snowfall detection based on visible wavelength surface reflectance changes would maximize the detection of the transition from snow-free to snowy ground. However, because visible light penetrates into the snowpack, detection of depositions of thin new snow layers would be obfuscated by reflectance contributions from older sub-surface snow layers, decreasing the detectable pre/post-snowfall reflectance difference. Therefore, we decided to use a parameter connected to the snow grain size, namely the specific surface area of snow (SSA, $\text{m}^2/\text{kg}$). SSA is calculated from the surface reflectance values measured at NIR wavelengths, where limited snowpack penetration enhances the surface layer contribution to the detected reflectance. The SSA estimation is based on Kokhanovsky et al. (2021). The main function is:

$$R = R_0 \cdot (e^{-\sqrt{\alpha L}})^f, \tag{4}$$

where, in our case, $R$ is surface reflectance from S2 band 9, $R_0$ is snow reflectance without absorption and is set as 0.99, $\alpha$ is the bulk absorption coefficient of ice defined for S2 band 9, $f$ is an angular function, dependent on sun zenith angle and instrument viewing angle, and $L$ is an effective light absorption path related to the snow specific surface area that we want to solve. More details can be found in Kokhanovsky et al. (2019a) and Kokhanovsky et al. (2021). The SSA values are then obtained by using the relationship SSA$= q/L$, where $q = 1.047$ $\text{m}^3/\text{kg}$ (Kokhanovsky et al. (2023)) and $L$ is obtained from Eq. (4). We calculate SSA values using surface reflectance values with and without forest correction implemented, resulting in

two SSA data sets. Also, because SSA values are calculated, not measured, we decided to limit the SSA values to a maximum of 160 m$^2$/kg (Gallet et al. (2009)). This is applied to both SSA data sets.

Because of metamorphism, snow grains grow larger as snow ages. This means that surface area decreases compared to fresh snow, that is, fresh snow increases SSA values, and conversely, older snow (no new snow) decreases SSA values. Therefore, we can detect possible snowfall events by calculating differences between two SSA values.

The uncertainty for SSA is determined by bootstrapping. We randomly choose (with replacement) 10000 data points, and we calculate SSA values using S2 channel 9 surface reflectance values with and without uncertainty included (modified data and reference data, respectively). Uncertainty for surface reflectance values is drawn randomly from the normal distribution $\epsilon \sim \mathcal{N}(0, 0.03^2)$. This bootstrapping is run 1000 times, and it resulted in mean uncertainty of 2.7 m$^2$/kg to SSA values without forest correction implemented, and 15.0 m$^2$/kg to SSA values with forest correction.

### 3.4 Detection threshold of snowfall-induced reflectance changes in S2 imagery

The determination of what amount of snowfall is accepted as a precipitation event in our study is not a straightforward task. The change in snow reflectivity depends on both the amount of fresh snow and its optical properties, and the associated change should be greater than the typical uncertainty in retrieved S2 surface reflectances. To explore the question, we simulated snow albedo changes resulting from fresh snowfall on top of existing old snow with the Two-streAm Radiative TransfEr in Snow (TARTES) snow model (Libois et al. (2013)). Prescribing an optically semi-infinite old snow cover (SSA set as 19 m$^2$/kg), we calculated the diffuse snow albedo change over the S2 B9 band from snowfalls depositing 0.05...15 cm of fresh snow with SSA of 40, 50, or 65 m$^2$/kg.

The S2 surface reflectance products have an uncertainty requirement of 5% (Gascon et al. (2017)), which translates to approximately 0.03–0.05 surface reflectance given typical snow reflectances in the B9 band. Accordingly, the TARTES simulation results (Figure 3) show that there needs to be at least 1 cm of snowfall to ensure detectability given the observational uncertainty. To change that to accumulated snowfall, we need to change it based on the snow-rain ratio, which is sensitive to temperature (e.g. National Centers for Environmental Information (2021)). The median value of all in situ temperature observations from March 2018 from the area of Luosto radar is -9°C and therefore, the 1 cm of snow is changed using a 1:20 ratio, leading to the minimum accumulated snowrate sum between observations (either SSA or SWE) for detectable snowfall to be 0.5 mm.

## 4 Results

### 4.1 SSA-based classification

Differences between SSA values can be calculated either tile-wise or pixel-wise. In a tile-wise approach, the whole tile is compared pixel by pixel to the next available tile, leaving missing pixels in either tile empty, leading to the time difference between pixels in two tiles being the same. In a pixel-wise approach, one pixel would be compared to the next available pixel,

regardless of the tile in which it is located. This leads to the diverse time differences between pixels in two tiles. The advantage of the pixel-wise approach is its larger number of data points to be used for analyses, but we decided to use the tile-wise approach because the results are easier to interpret.

The limit for SSA difference was set to either zero or SSA uncertainty, leading to four different classification cases: classification for SSA differences without forest correction step and with change limit either zero or 2.7 m$^2$/kg (marked as SSA$_0$ and SSA$_u$, respectively), or classification for SSA differences with forest correction step and with change limit either zero (SSA$_{f0}$) or 15.0 m$^2$/kg (SSA$_{fu}$). We also combined classification results from SSA$_0$ and SSA$_u$ (marked as SSA$_{comb}$). In this combination classification, a pixel was classified as snowfall or no-snowfall, if both classifications agreed. If not, then the classification value was omitted. The confusion matrices and statistics for each classification case are shown in Table 1 and in Table 2, respectively.

From the cases SSA$_0$, SSA$_u$, SSA$_{f0}$, and SSA$_{fu}$, the SSA$_{f0}$ (SSA differences with forest correction step and with change limit set as zero) have the highest accuracy (78%). The SSA$_{f0}$ classification detects 88% of all radar-based snowfall occurrences (recall), and it also classifies 77% of snowfall cases correctly (precision). For situations without snowfall, the percentages are 63% and 79%, respectively. On the other hand, SSA$_0$ (SSA differences without forest correction step and with change limit set as zero) yields a better recall value (71% ) for no-snowfall cases. We can also see that including uncertainty as a change limit decreases significantly the amount of pixels but does not yield better results. Combining the classification results from SSA$_0$ and SSA$_u$ increases all statistics. Accuracy is 83%, and recall value for snowfall is above 90%. The disadvantage is the decrease in coverage (around 10 000 fewer pixels compared to the SSA$_0$ and SSA$_u$). Regardless of the decrease in the number of pixels in the combined classification, the statistical values are still comparable. We used bootstrapping method to generate random samples from the classification results. We generated separately 1000 samples with a sample size being 10000 from SSA$_0$, SSA$_{f0}$ and their combined results, and calculated statistical values (recalls, precisions, f1 scores, and accuracies) for each 10000-sized sample. Then we took the mean values of those 1000 statistical values. These bootstrapped recalls, precisions, f1 scores, and accuracies are almost the same as the result in Table 2, leading to the difference between total populations of SSA$_0$ and SSA$_{f0}$, and their combined classification being insignificant.

Examples of classifications are shown in Figures 4 and 5. In Figure 4, which is an example of snowfall situations (collected from classifications using tiles with the same dates, 15 March and 20 March), the importance of forest correction can be seen. Large areas are classified incorrectly when the forest correction step is excluded (panel b), even though one tile has some challenges in the forest-corrected classification results (panel c), the reason for that is not clear. Majority of misclassifications in the results without forest correction (panel b) are most likely due to the canopy interception of snow not happening. As the interceptions do not only depend on forest canopy cover, but also for example air temperature (Miller (1964)), wind (McNay et al. (1988)), and topography (D'Eon (2004)), it is not a straightforward task to determine why in those particular areas the canopy interception did not happen. The forest correction is therefore important step, as it corrects these missed canopy interception cases (panel c). An example of situations without snowfall is shown in Figure 5. In this particular tile, the data without the forest correction step (panel b) yields better classification results than when forest correction is included (panel c).

The combined results (panel d in both Figures 4 and 5) look more similar to the radar-based snowfall information (panel a in both Figures)).

## 4.2    SWE-based classification

In addition to the optical-based SSA classification, we also compared daily SWE differences with radar-based reference data to see how well changes in SWE can detect snowfall in spring. The satellite-based SWE retrievals are primarily based on snow cover microwave emission detection using 19 and 37 GHz wavelengths and, therefore, the retrievals are insensitive to variations in solar illumination, cloudiness, and most weather conditions. This leads to better spatial and temporal coverage relative to optical satellite measurements, although at the expense of spatial resolution which is considerably coarser for passive microwave radiometers.

The daily time series of satellite-based SWE classification is shown in Figure 6. A notable daily variability exists in the classification, with high consistency between methods on some days and large discrepancies on others. The accuracy is 53% (Table 2) which is lower than any of the SSA classification accuracies. The SWE-based classification detects 64% of all the radar-based snowfall occurrences (recall) and correctly classifies 63% of snowfall cases (precision). For situations without snowfall, the percentages are 34% and 36%, respectively.

In addition to satellite-based SWE, we also investigated daily SWE differences from ERA5L with radar-based reference data to ensure the usability of this method. The ERA5L-based classification shows a notably higher accuracy (78%; Table 2) compared to the satellite-based classification. Time series (Figure 7) show that the classification is accurate especially in the first half of the month. The number of misclassifications increase towards the end of the month, but is still relatively high compared to the satellite-based classification (Figure 6). The satellite-based SWE classification is more accurate in detecting snowfall than situations without snowfall. For ERA5L, such difference is not evident. The ERA5L-based classification detects 77% of all the radar-based snowfall occurrences (recall) and correctly classifies 86% of snowfall cases (precision). For situations without snowfall, the percentages are 80% and 68%, respectively.

Classification examples for two different days are shown in Figures 8 and 9. From Figure 8, we can observe that the radar detects snowfall in approximately one third of the study area, while satellite-based SWE classification detects snowfall in an area much larger than the radar data. ERA5L, in turn, is highly consistent with the radar data. Furthermore, Figure 9 shows that both SnowCCI and ERA5L fail to detect all the spatial variability in snowfall. The original resolutions of radar and SWE datasets differ considerably, thus, possibly leading to uncertainty in the classification.

We additionally investigated how elevation and forest cover fraction affect the classification (Figures 10 and 11). Overall, ERA5L shows higher accuracy than the SnowCCI, which was already evident from Table 2 and Figures 6 and 7. Also, SnowCCI is better in detecting snowfall events than situation without snowfall, while such difference is not apparent in ERA5L. Figures 10 and 11 show that the overall accuracy (dark blue line) does not exhibit notable dependency on elevation or forest cover. However, SnowCCI is able to detect situations without snowfall more accurately with higher elevation and less dense forest.

## 5 Summary and discussion

Over the high latitudes of the Northern Hemisphere, precipitation in the form of snow is responsible for creating and growing seasonal snowpacks. Current atmospheric reanalyses and direct satellite-based precipitation observations suffer from high variability or limited spatiotemporal coverage and thus are not ideal for detecting high-latitude snowfall events. Therefore, we decided to utilize satellite observations measured at the optical and microwave wavelength ranges. Using optical measurement-based specific surface area of snow (SSA) and microwave-based snow water equivalent (SWE), we were able to detect snowfalls

with high accuracy, but cases without snowfall turned out to be more difficult to classify.

We used radar-based snowfall information as the reference data, i.e. "ground truth". Due to the wind drift, we needed to do a wind adjustment to processed snowrate values. As tree height is around 10 m in Northern Finland (FMI (2022)), we adjusted DEM slightly for the wind adjustment by adding 10 m to all forested pixels. Changing this value to either 0 m or 30 m did not affect the classification results. Nevertheless, we decided to use a tree height of 10 m for the completeness of the study. Another

325 possible parameter of wind adjustment to affect the classification results is snowfall speed. The used 1 m/s is a typical value for snowfall speed (Lauri (2010), Ishizaka et al. (2016), Vázquez-Martín et al. (2021)). Lauri (2010) also states that fall speed have a spectrum width of about 0.3 m/s. Therefore, we did additional simulations with wind speed of either 0.7 m/s or 1.3 m/s and compared classification results to the classifications we obtained by using wind speed 1 m/s. These changes did not change the classification results. Because our study area is a circle with a 100 km radius around the radar site and the spatial resolution is

330 either 1 km or 5 km, the distance between the detection height of the snowfall and the ground is not long enough to different fall speeds to affect where snow falls at the grid cell scale. With a higher distance from the radar (> 100 km), the distance between snow detection height and ground increases, and hence the probability of snow falling to different pixels increases. Therefore, uncertainties due to the chosen constant tree height and fall speed do not cause almost any uncertainty about the actual location of the snowfall. Regardless, in its entirety the wind adjustment is important part of the retrieval process. We additionally did the

335 classifications with SSA values and 1 km resolution data by using snowrate values without wind adjustment included. Almost all statistical values (recalls, precisions, f1 scores, and accuracies) in all cases decreased compared to the classification values acquired with wind-adjusted snowrate values. Especially accuracies in each case decreased around 0.03, indicating that wind adjustment is a necessary step for maximizing accuracy.

Using optical-based satellite measurements to detect snowfalls is not a straightforward task, and that may be the reason it has

340 not been used very widely. We considered multiple different optical satellite products to be used in this study, but with almost all of them, we had similar challenges: the resolution was not fine enough to classify snowfall correctly, Bidirectional reflectance distribution function (BRDF) over snow proved to be difficult to implement, and also densely forested areas combined with the coarse resolution (difficult to differentiate between forested areas and open spaces) made it challenging to detect new snow atop older one. Sentinel-2 MSI measurements turn out to be the most suitable data set to use, due to its very fine spatial resolution

(10–60 m) at near-nadir viewing angles, good radiometric precision, and easy accessibility.

Previously, snowfalls have been linked to the increased SSA values (Libois et al. (2015), Kokhanovsky et al. (2019b)), and in our study, we use this connection conversely to detect snowfalls with good results: from 77% to 91% of snowfall cases are

classified correctly (depending on the used data set) when comparing to the reference data. Some of the misclassifications (both snow and no-snow cases) are due to the remaining clouds and cloud shadows, as it is typically difficult to identify correctly clouds over bright snow cover. Smaller-scale misclassifications are mostly due to the higher temperatures at the end of March 2018 (the study month and year) and the effects of wind. Wind sublimates and fragments snow crystals (Domine et al. (2009)), causing SSA values to increase without snowfall. In this study, we assume that the wind effect on the SSA values is minor due to the forested areas, i.e. limited amount of open spaces. Also, studies have shown that albedo begins to decrease due to snow metamorphism when the air temperature rises above -5°C (Kouki et al. (2019)), which can have a slight impact on the SSA-based classification.

The microwave-based SWE was chosen for this study because snowfall is assumed to directly increase SWE over an area. Also, contrary to optical measurements, microwave-based observations do not require sunlight and are not affected by clouds. Therefore, SWE estimates are available during the entire winter season. Currently, the SnowCCI is the only satellite-based SWE product covering the entire NH and several decades. The most recent version of the SnowCCI SWE product (version 2) is a well-suited product for this research because the seasonal evolution of SWE is accurately described in the product compared to the older SnowCCI version 1 (Mortimer et al. (2020)). In addition to the satellite-based SWE, we also included SWE data from ERA5L reanalysis to support the analysis. SWE-based classifications, surprisingly, were not as good as SSA-based classifications, only around 64% (SnowCCI) and around 77% (ERA5L) of snowfall cases were classified correctly compared to the reference data. The original resolution of the SWE data is 0.1° (about 10 km) for SnowCCI and 9 km for ERA5L which are notably coarser than the resolution of the reference data. Therefore, it is likely that the different spatial resolutions of the compared products reduce the accuracy of the classification. Also, the analysis revealed that the SSA-based classification shows higher classification accuracy than either of the SWE-based classifications. The spatial resolution of the S2 data used in the SSA-based comparison is 60 m, which is considerably finer than the resolutions of the SWE data. This suggests that the spatial resolution of the satellite data affects the classification, i.e a coarse resolution reduces the accuracy.

In contrast, the classification of no-snowfall cases turns out to be a more challenging task for satellite-based data. The aging of snow grows snow grains on the snowpack surface more slowly than snowfall increases them. Therefore, the decrease in SSA values may be undetected due to the measurement uncertainties, causing misclassifications. Still, from almost 60% to over 70% no-snowfall cases are correctly classified compared to the reference data. Based on the results, satellite-based SWE is mainly sensitive to snowfall, as only 34% of no-snowfall cases are correctly classified. The higher temperatures at the end of March can also impact SWE values, causing misclassifications. The analyses also show higher statistical values for cases with and without snowfall for classifications using SWE from ERA5L compared to SWE from SnowCCI. This disagreement is mostly due to the information about the snowfall (or lack of it) being included to the ERA5L-based SWE calculations (ECMWF (2016)). This leads to the conclusion that the satellite-based SWE can be used to detect snowfall events, but using it to classify no-snowfall cases is not recommended.

In the future, we need to use more data and cover larger areas, as well as study the sensitivity of the chosen resolution to the results to be able to achieve more reliable classification results. Also, in the future, the goal is to apply these classifications for the entire Arctic and a longer time period. Using 1 km or 5 km resolutions is too fine when covering the whole Arctic, but one

idea could be to do first-stage classifications using finer resolution data and then perform analyses to larger areas using coarser resolution (e.g. 10 km).

This proof-of-concept study was limited in the spatiotemporal domain, considering only March 2018 over an area of approximately 31400 sq. km in Northern Finland. Nevertheless, the indirect snowfall detection from both optical and microwave satellite observations yielded encouraging results. Correct classification of no snowfall proved more challenging, as discussed above, yet further improvements in the classification remain possible and could yield a robust snowfall detection method applicable for large remote regions where e.g. weather radar observations are not available. Naturally, questions regarding the

generalization of the method trained with weather radar data from Finland to other regions and the validation of the ensuing estimates would then need to be explored in detail.

## Appendix A: Wind adjustment algorithm

The wind drift adjustment method is from Lauri (2010), and here we provide outline of the algorithm. The basic idea is to find where the possible snowrate value would have come from to the surface. The falling speed of snow, $w$, is assumed to be 1 m/s

(Lauri (2010), Ishizaka et al. (2016), Vázquez-Martín et al. (2021)). The process in our case is as follows:

1. Set a blank matrix $S \in \mathbb{R}^{\text{rows}\times\text{cols}\times\text{time}}$

2. For each pixel $(r, c, t)$ in $S$:

   (a) set time $t$ as $t_0$

   (b) fetch wind data vectors $z_i$ $u_i$ and $v_i$ for the pixel $(r, c)$

(c) determine upper $(j_u)$ bound of geopotential height which is based on radar minimum height at location $(r, c)$

   (d) determine lower $(j_l)$ bound of geopotential height which is based on $\text{DEM}_{\text{adj}}$ (adjusted DEM) at location $(r, c)$

   (e) calculate pixel movements $r_m$ and $c_m$:

$$r_m = \sum_{j=j_l}^{j_u} \left( \frac{z_j - z_{j+1}}{w} \cdot \frac{v_j + v_{j+1}}{2} \right)$$

$$c_m = \sum_{j=j_l}^{j_u} \left( \frac{z_j - z_{j+1}}{w} \cdot \frac{u_j + u_{j+1}}{2} \right) \tag{A1}$$

   (f) add pixel movements to pixel location $(r, c)$:

$$r_1 = (r + 0.5) + (-1)r_m$$

$$c_1 = (c + 0.5) + c_m \tag{A2}$$

   (g) calculate time movement in minutes:

$$t_m = ((z_{j_u} - z_{j_l}) \cdot w)/60 \tag{A3}$$

(h) determine adjusted time step:

$$t_1 = t_0 - \lfloor t_m/10 \rceil \tag{A4}$$

(i) insert snowrate value to $\boldsymbol{S}$:

$\boldsymbol{S}(r,c,t)$ = snowrate value at pixel location $(r_1, c_1)$ and time step $t_1$

In the step 2f, the $(-1)$ is used to change the direction of the component $v$ from south-to-north to north-to-south. We also assume that each observation is located in the center of the pixel, and therefore we need to add 0.5 km to each movement (step 2f). We divide the movements by 1000 as to change them from meters to kilometers even though it is not denoted in the

415 formulas. For 5 km resolution data, the row and column movements are divided by 5 and rounded. Time movement (step 2g) is to find how long it will take for the snow to actually fall to the ground, and whether the observed value should be taken from some previous time layer (step 2h). The marking $\lfloor x \rceil$ indicates rounding.

We do not change the $z_i, u_i$ and $v_i$ data vectors within one iteration after the initial setting, i.e. we use the values set in step 2b. The ERA5 data have a coarse resolution, and therefore the values do not change a lot around the Luosto area within each

420 layer.

*Author contributions.* AR is responsible for acquiring the funding and is also behind the conceptualization of research questions presented in this manuscript. AR, as the PI of the funding project, was supervising the research work. All authors took part for collecting and processing data. EJ developed methodology for analyses. EJ and KK carried out analyses and investigated the obtained results. EJ led the preparation of manuscript with contributions from all co-authors. All authors reviewed and edited the manuscript.

*Competing interests.* The authors declare that they have no conflict of interest.

*Acknowledgements.* This work was financially supported by the Research Council of Finland in the project SNOCAP (341845). The authors would like to thank the Copernicus Data Space Ecosystem and the European Space Agency for providing satellite data used in this study, the Finnish Meteorological Institute for providing weather radar data, and the National Land Survey of Finland and the Natural Resources Institute Finland for providing auxiliary data used in this study. The authors would like to thank Markus Peura, Annakaisa von Lerber, Harri

Hohti, and Niilo Kalakoski (Finnish Meteorological Institute) for helpful discussions.

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

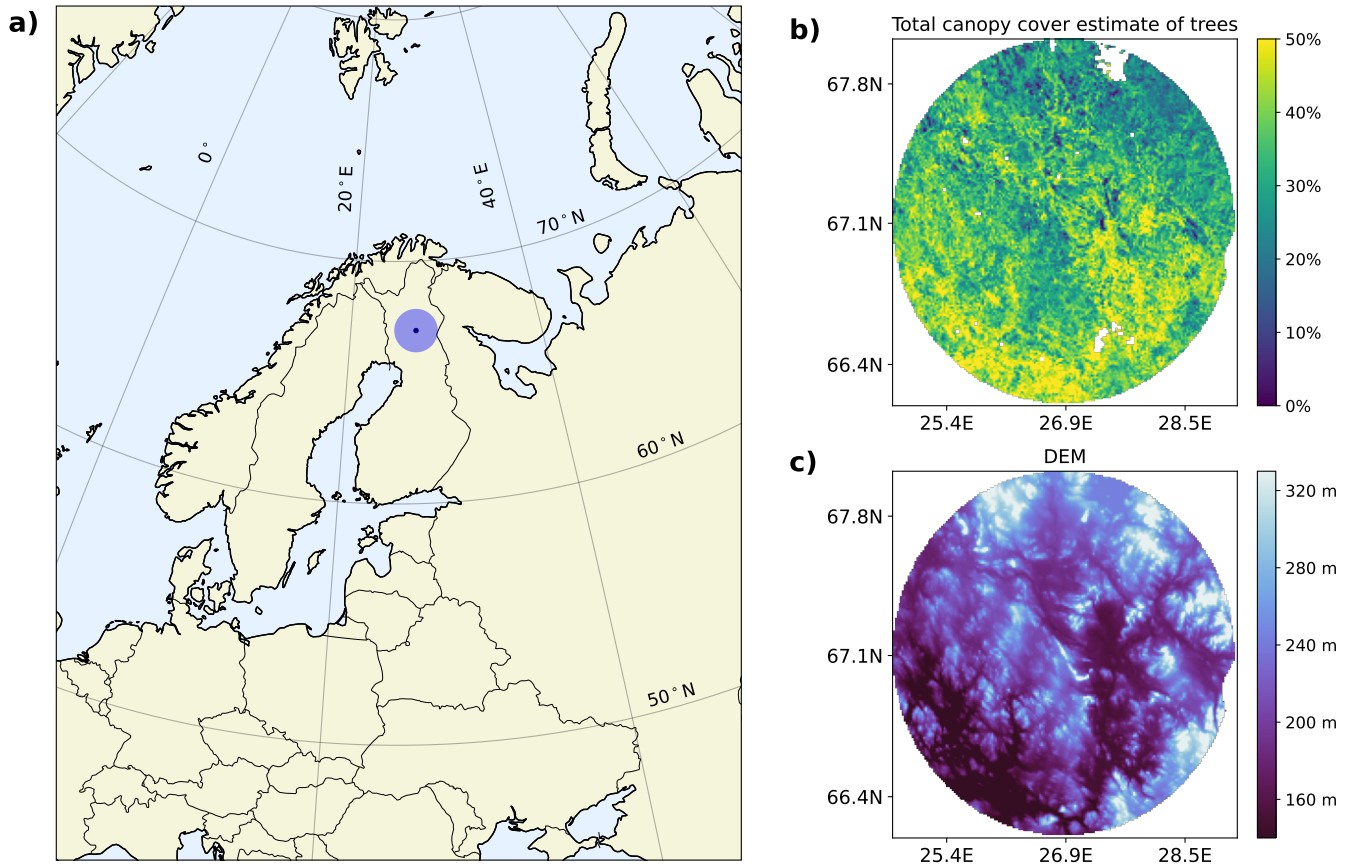

**Figure 1.** Location of the study area (panel a), and total canopy estimate of trees (panel b) and digital elevation map (panel c) for 100 km radius around the Luosto radar.

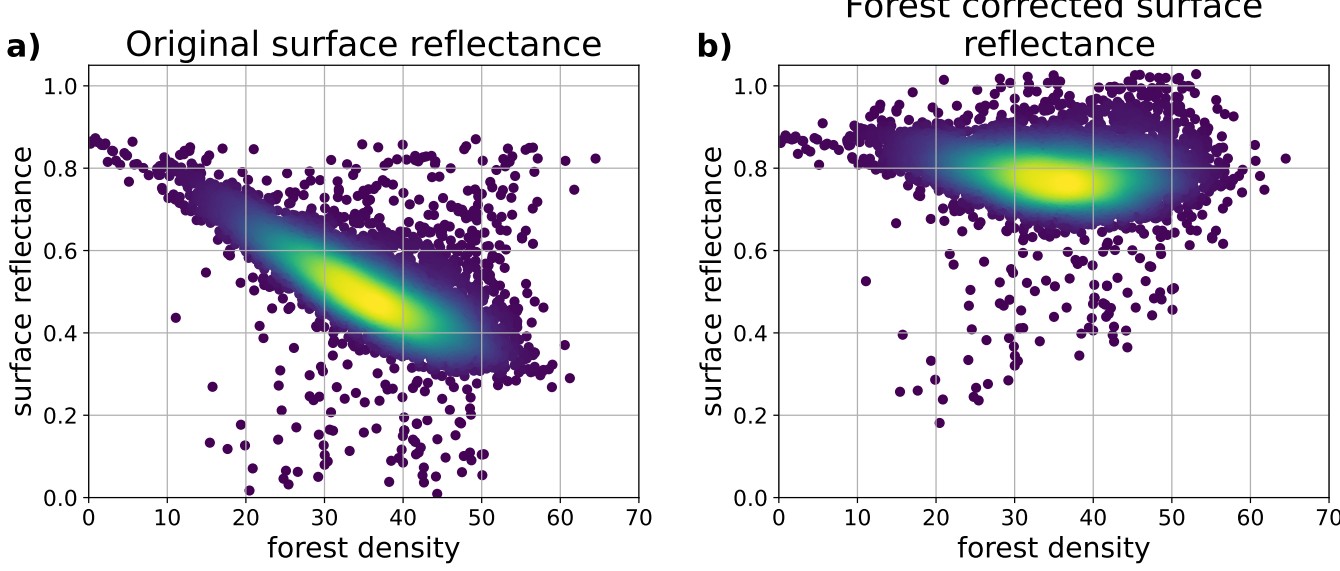

**Figure 2.** An example of surface reflectance values from Sentinel-2 band 9 in relation to forest density from 15 March 2018. Original surface reflectance values are shown in panel a, and forest corrected surface reflectance values are shown in panel b.

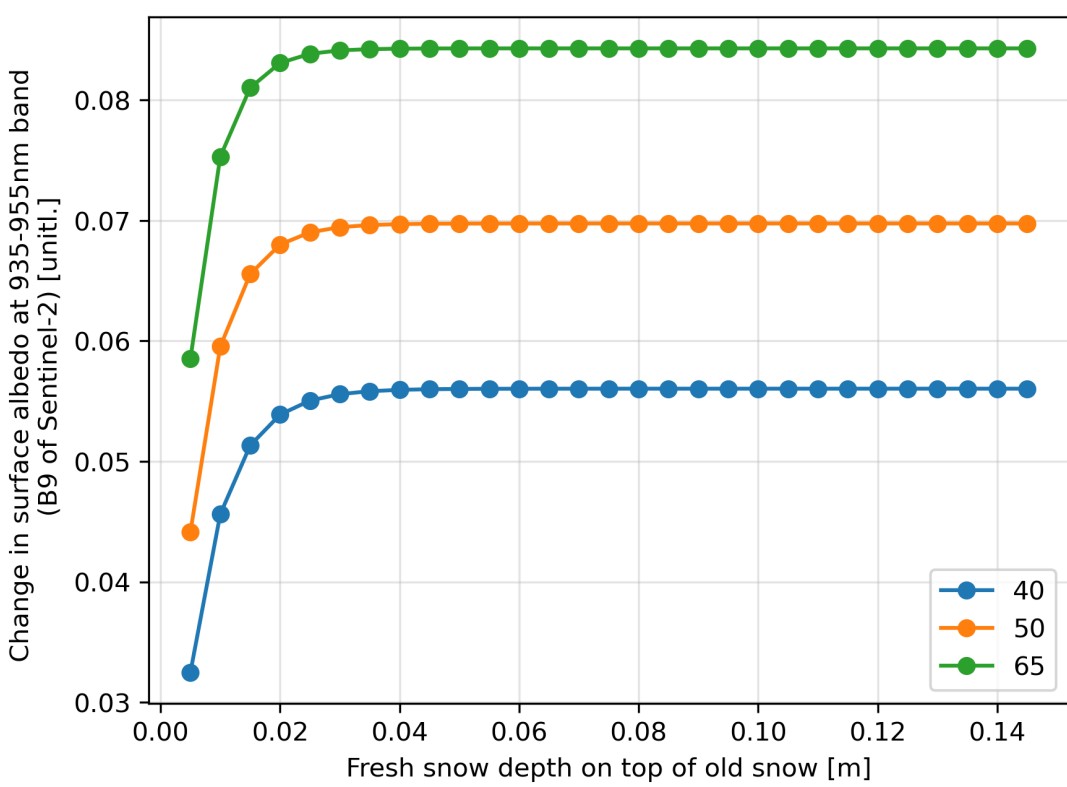

**Figure 3.** Change in white-sky albedo simulated by TARTES as a function of fresh snow deposition of varying SSA, 40 (blue), 50 (orange) or 65 (green) m$^2$/kg. The underlying old snow is prescribed as optically semi-infinite with SSA of 19 m$^2$/kg.

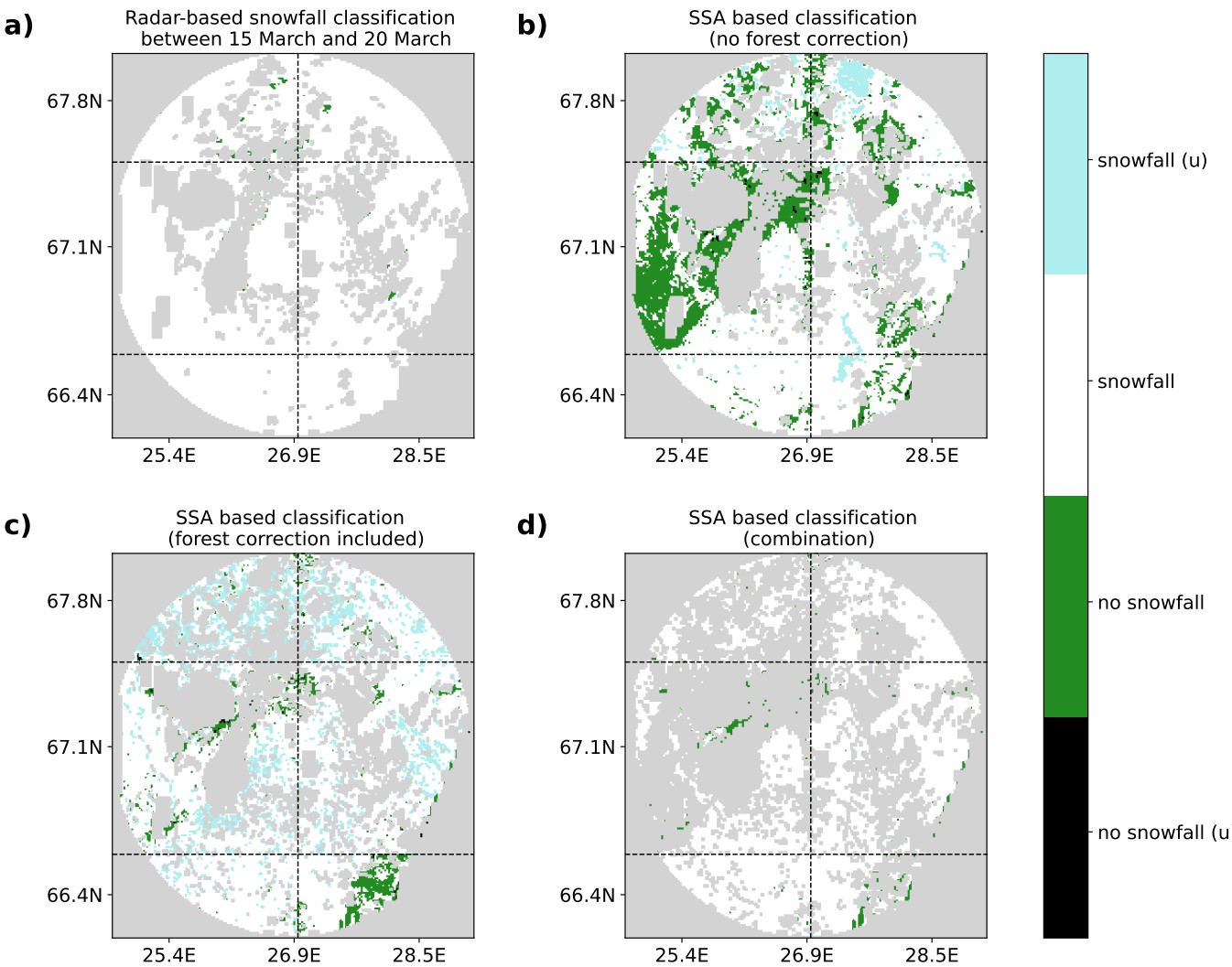

**Figure 4.** Example figure of different classifications for observations between 15 March and 20 March. White color indicates snowfall, light blue snowfall with uncertainty, green color indicates for no snowfall, and black for no snowfall with uncertainty. Grey color is for missing or omitted values. Dashed lines indicate the S2 tile borders. Panel a) classification of snowrate data from radar; panel b) classification of SSA differences without forest correction; panel c) classification of SSA difference with forest correction; panel d) combination of SSA classifications.

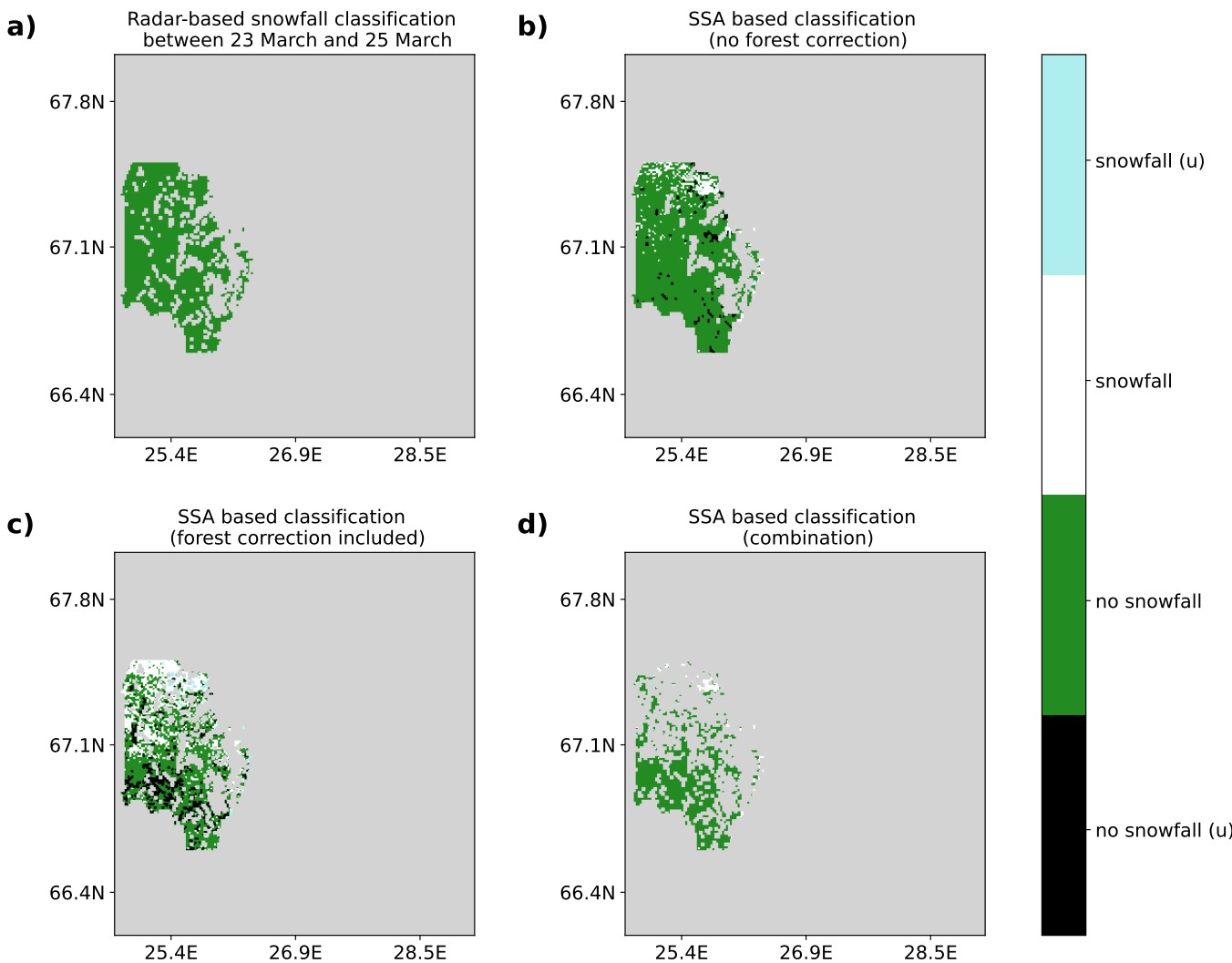

**Figure 5.** Same as Figure 4, but with dates 23 March and 25 March.

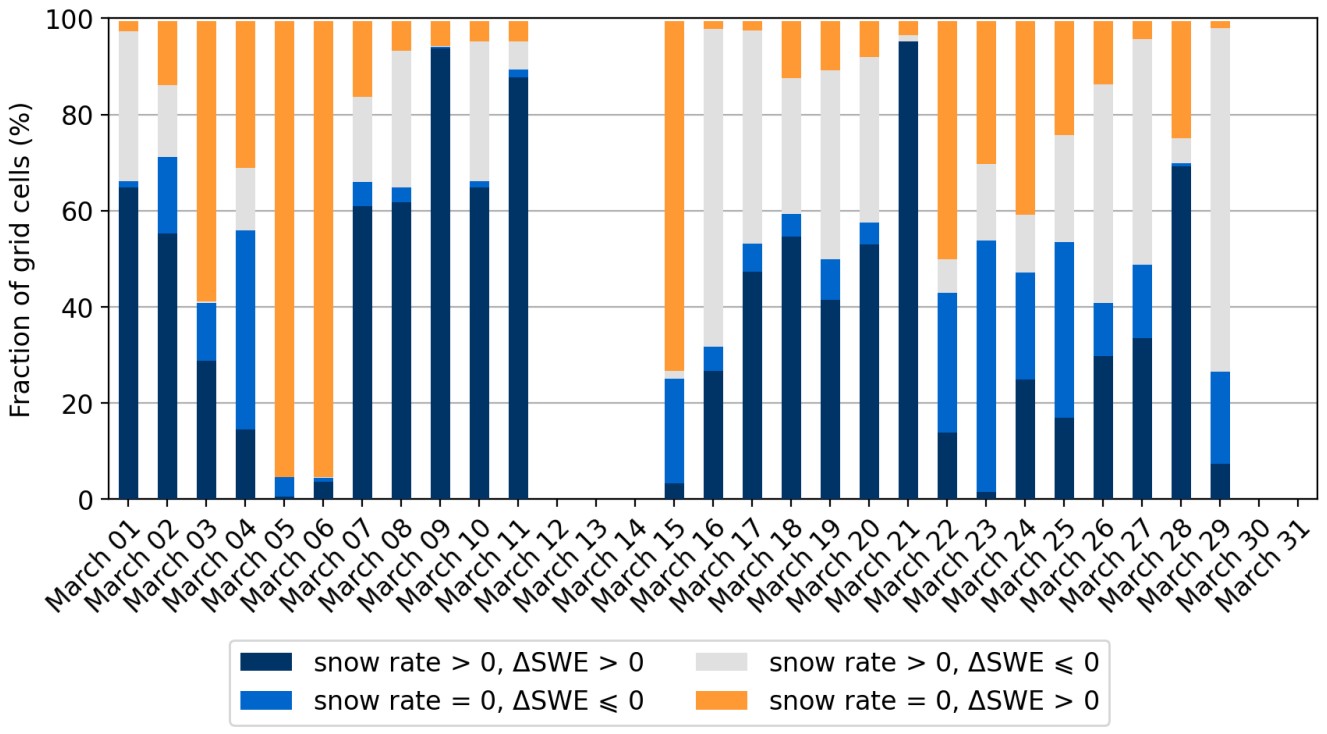

**Figure 6.** Daily time series of SnowCCI SWE-based classifications. Blue colors (dark blue and lighter one) indicate agreement between SWE data and radar-based snowfall information. The light grey and orange colors indicate disagreement between SWE and radar snowfall information. The gaps in the time series are due to missing values in the SnowCCI data.

.

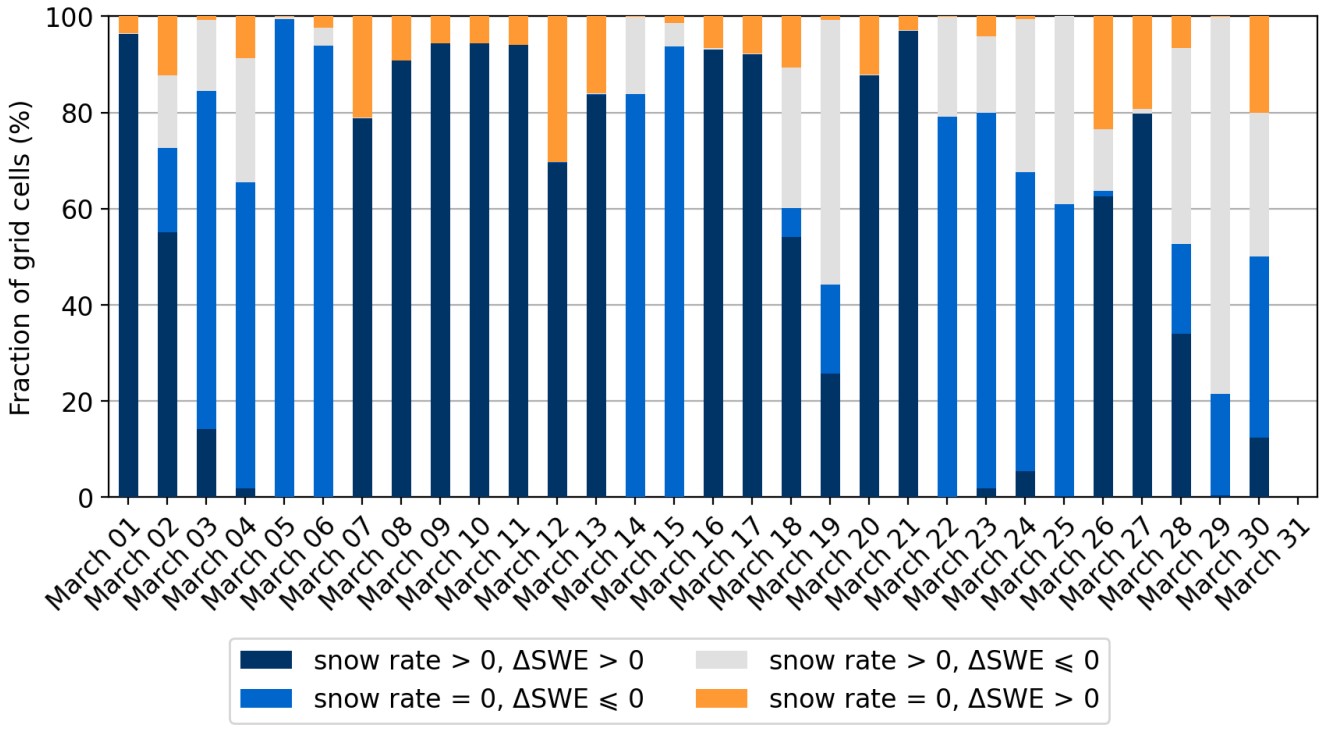

**Figure 7.** Daily time series of ERA5L SWE-based classifications. Blue colors (dark blue and lighter one) indicate agreement between SWE data and radar-based snowfall information. The light grey and orange colors indicate disagreement between SWE and radar snowfall information.

.

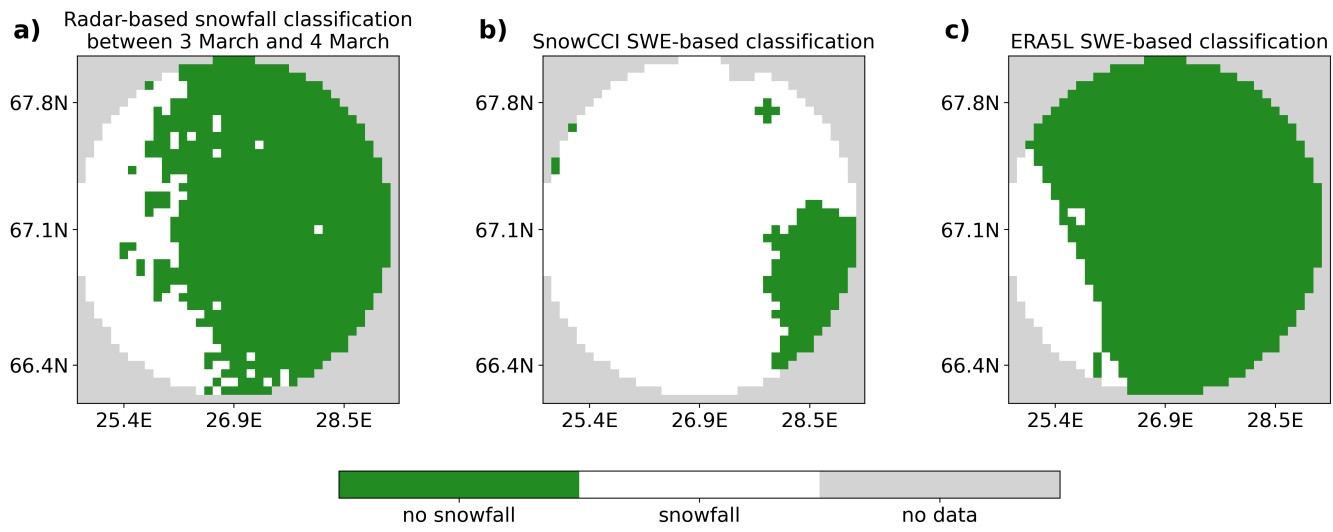

**Figure 8.** Example figure of radar-based and SWE-based classifications for observations between 3 March and 4 March.

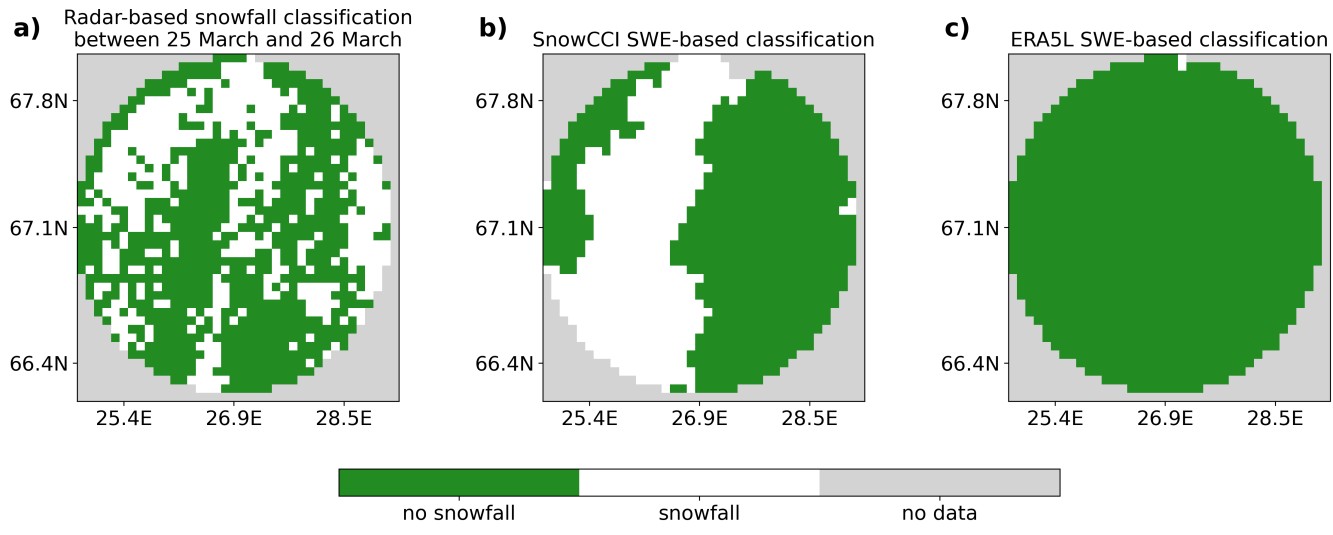

**Figure 9.** Example figure of radar-based and SWE-based classifications for observations between 25 March and 26 March.

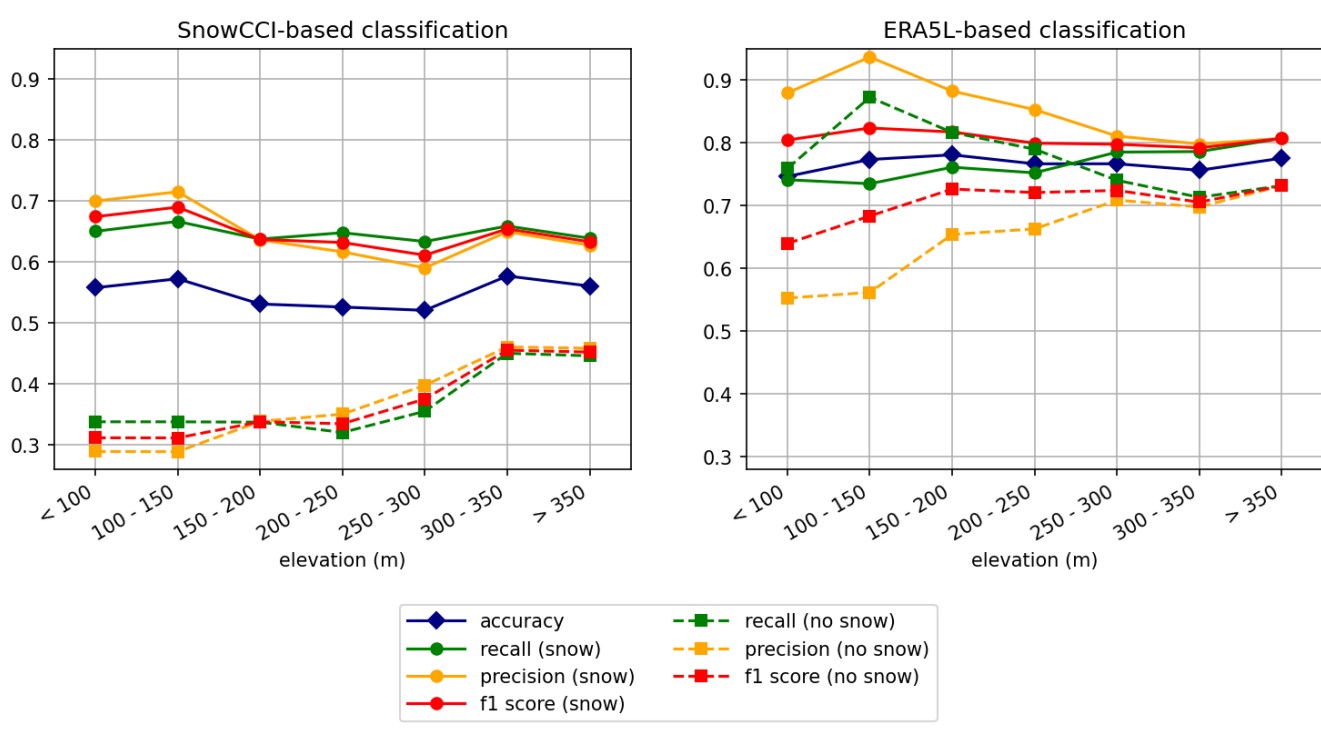

**Figure 10.** Dependency of the statistics on elevation for SnowCCI and ERA5L-based classification.

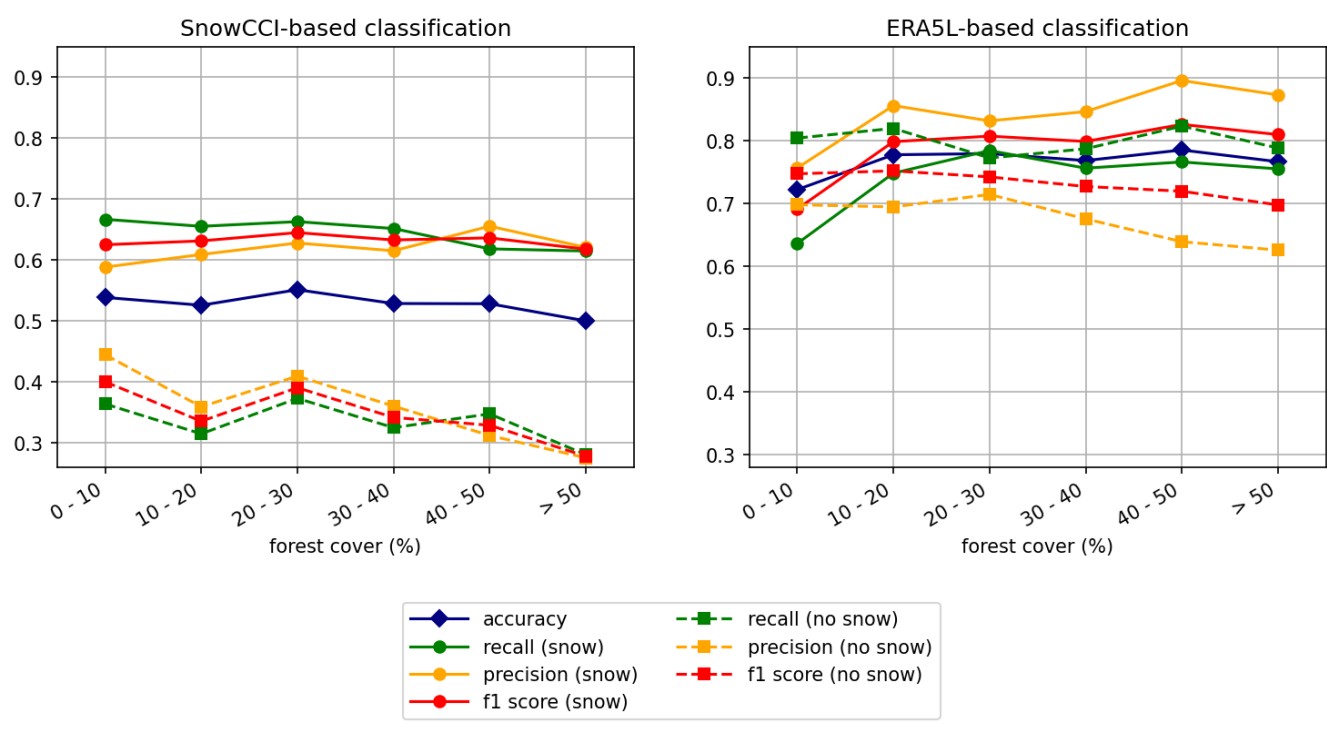

**Figure 11.** Dependency of the statistics on forest cover for SnowCCI and ERA5L-based classification

**Table 1.** Confusion matrices for five different classification cases based on SSA and for one classification based on SWE. The unit of measure for change limits for SSA-based classifications are $m^2/kg$.

| | limit | snow | no snow | total |
|---|---|---|---|---|
| $SSA_0$ | SSA diff $> 0$ | 22156 | 5896 | |
| | SSA diff $< 0$ | 6461 | 14517 | 49030 |
| $SSA_{f0}$ | SSA diff $> 0$ | 24070 | 7109 | |
| | SSA diff $< 0$ | 3176 | 12230 | 46585 |
| $SSA_u$ | SSA diff $> 2.7$ | 1112 | 550 | |
| | SSA diff $< -2.7$ | 251 | 800 | 2713 |
| $SSA_{fu}$ | SSA diff $> 15.0$ | 3932 | 1291 | |
| | SSA diff $< -15.0$ | 770 | 1977 | 7970 |
| $SSA_{comb}$ | $(SSA_0 = 1)$ & $(SSA_{f0} = 1)$ | 19927 | 4131 | |
| | $(SSA_0 = 0)$ & $(SSA_{f0} = 0)$ | 2083 | 10868 | 37009 |
| SnowCCI SWE | SWE diff $> 0$ | 12362 | 7323 | |
| | SWE diff $< 0$ | 6823 | 3782 | 30290 |
| ERA5L SWE | SWE diff $> 0$ | 16625 | 2779 | |
| | SWE diff $< 0$ | 5117 | 10609 | 35130 |

**Table 2.** Statistics from the confusion matrices in Table 1.

|  | recall (SNOW) | recall (no SNOW) | precision (SNOW) | precision (no SNOW) | f1 score (SNOW) | f1 score (no SNOW) | accuracy |
|---|---|---|---|---|---|---|---|
| $SSA_0$ | 0.77 | 0.71 | 0.79 | 0.69 | 0.78 | 0.70 | 0.75 |
| $SSA_{f0}$ | 0.88 | 0.63 | 0.77 | 0.79 | 0.82 | 0.70 | 0.78 |
| $SSA_u$ | 0.82 | 0.59 | 0.67 | 0.76 | 0.74 | 0.67 | 0.70 |
| $SSA_{fu}$ | 0.84 | 0.60 | 0.75 | 0.72 | 0.79 | 0.66 | 0.74 |
| $SSA_{comb}$ | 0.91 | 0.72 | 0.83 | 0.84 | 0.87 | 0.78 | 0.83 |
| SnowCCI SWE | 0.64 | 0.34 | 0.63 | 0.36 | 0.64 | 0.35 | 0.53 |
| ERA5L SWE | 0.77 | 0.80 | 0.86 | 0.68 | 0.81 | 0.68 | 0.78 |