# Peer review of "Detecting Snowfall Events over the Arctic Using Optical and Microwave Satellite Measurements"

_Hydrology and Earth System Sciences, 2023_

## Author Comment (AC1)

**Answers to referee #1**

We thank the referee for their thoughtful comments on our manuscript. Answers to the referee's comments are below. Our answers are marked with blue color.

The authors use two different satellite data products (one optical and one microwave) to estimate snowfall occurrence near one weather radar site in Finland in March 2018. This is a proof-of-concept study that demonstrates feasibility of the techniques for detecting snowfall occurrence, although both techniques struggle much more to detect non-occurrence.

It is always somewhat difficult to know whether to recommend publication of proof-of-concept studies, since they invariably raise as many questions as they answer and demand further related work to realize their scientific potential. This article is no exception: it is well-written, scientifically thoughtful and well-reasoned, and the results are clear and intriguing. Yet, the sample and reference data are highly limited in time and space, and seemingly important caveats and limitations are highlighted but not investigated in sufficient detail. My instinct is that the study requires a bit more fleshing-out before it can be published as a standalone contribution.

L63: Metamorphism that affects snow albedo is still possible at temperatures below freezing (e.g., Qu and Hall 2007). How does this affect your assumptions?

Yes, that is true. However, studies have shown that snow metamorphism only begins to affect albedo when temperature rises above -5°C (e.g. Pirazzini, 2004, Pirazzini et al., 2006, Kouki et al., 2019). We checked the temperatures of the closest weather station (Sodankylä weather station, which is in our study area), and it shows that temperatures remain below -5°C most of the chosen time period, only just at the end of the March 2018 temperatures rise above that limit. Given the modest dry snow metamorphism-induced reflectivity changes in snow below this limit, we therefore consider that we do not need to take snow metamorphosis into further account. We will modify this part of the manuscript.

L85: Consider adding ERA5-Land SWE as a reanalysis-based reference product. It should perform about as well as ESA-CCI SWE and has many advantages.

We will add the suggested dataset to the analysis.

L95: Typo of "Parameter"?

Yes, indeed, there is a typo, we will correct that.

L140: The wind adjustment section was somewhat brief, and the reader couldn't assess how important this correction is. How big of an effect comes from the wind adjustment? What is the

difference in snow rate before/after adjustment for wind? What is the sensitivity (if any) to assumptions about tree height and snow fall speed?

1. The tree height assumption of 10 m was based on the knowledge we have of Finnish boreal forests. We will add a citation to the online page: https://en.ilmatieteenlaitos.fi/ghg-sodankyla-forest to the Section "3.2 Wind adjusted snowrate data". On that online page, there is information about the forest around Sodankylä (which is close to the Luosto radar site), and the observed mean vegetation height is 12 m. We did additional simulations by using tree heights 0 m and 30 m, and neither of these changes affected the results. We will add more text about this to the Discussion section.

2. Our chosen constant snowfall speed of 1 m/s is a typical value for snowfall (Lauri (2010), Ishizaka et al. (2016), and Vázquez-Martín et al. (2021)). However, Lauri (2010) also stated that fall speed have a spectrum width of about 0.3 m/s. Therefore, we did additional simulations with a wind speed of either 0.7 m/s or 1.3 m/s and compared classification results to the classifications we obtained by using a wind speed of 1 m/s. It turns out, that with our area being a 100 km radius around the radar site, the snowfall speed with 0.3 m/s variation does not affect to the classification results. We assume, that if we had a larger area (radius > 100 km), then the snowfall detection height would be higher and the distance between snowfall and ground would be greater, leading to a higher probability of snowfall ending up in different pixels with varying snowfall speed. We will add more text about this to the Discussion section. We will also add additional citations about the chosen snowfall speed to the Appendix where we go through the wind adjustment.

3. Instead of comparing wind adjusted snowrate to the snowrate data without wind adjustment, we compared the classification results from using both of these data sets, because we are focusing on whether there is enough snow in the pixels instead of the actual amount of it. It turns out, that without wind adjustment, most of the statistical values (recalls, precisions, f1 scores, and accuracies) decrease, especially accuracy results decrease about 0.03 in all cases when focusing on SSA-based classifications. It is not much, but still a clear indication that wind adjustment is a necessary step. We will add a couple of sentences about this to the Discussion section.

L257: It is not immediately obvious from the maps what consistitutes a pixel and what is a tile. Therefore, we can't tell where the one tile is that has some challenges in panel 4c. Perhaps the authors could add a grid showing the layout of the tiles as a reference, say on Fig.1 or Fig.4?

Yes, that is true, there might be some confusion about tiles and pixels. We will add a grid to show the layout of the tiles in Figure 4.

L256: The areas of misclassification in Fig.4b without forest correction do not appear to be the areas of highest forest canopy cover (Fig.1b). How do the authors explain this apparent contradiction?

Yes, there are large areas of misclassified pixels over the area where the forest canopy cover is not especially high. The main reason for those misclassifications is the different canopy interceptions of snow. As the interception is not only dependent on forest canopy cover, but also for example air temperature (Miller (1964)), wind (McNay et al. (1988)), and even topography (D'Eon (2004)), it is not a straightforward task to determine why in those cases the canopy interception did not happen. We will add more text about the reason for these misclassifications to the manuscript.

L260: While the combined results show fewer misclassified pixels, they also have a considerable increase in missing or omitted values. So it is not completely straightforward to compare the performance of combined vs single-factor results. How is this difference in pixel number taken into account in the bulk metrics of Table 2?

Yes, it is true, that the combined results have fewer pixels than the original ones. The difference in pixel number is not considered when calculating statistical metrics from the results. We did bootstrapping (with replacement) tests with the data. We took 10 000 samples separately from classification cases $SSA_0$, $SSA_{f0}$, and $SSA_{comb}$ 1000 times, and calculated statistical values (recalls, precisions, accuracies, and f1 scores) using those 1000 iterations. Then we took mean values from each statistical value. These acquired results are almost the same as the statistical values in Table 2 for $SSA_0$ $SSA_{f0}$, and $SSA_{comb}$. Therefore, the statistical results from the combined classifications and the original ones are comparable. We will add more text about this to the manuscript.

L263: The section on SWE-based classification was incredibly brief. So much so that I found that it scarcely met the bar for demonstrating a proof-of-concept. What factor(s) are most important for explaining the (mis)classification? Presumably microwave emission is sensitive to the presence of surface water, topography, vegetation type/density, among other things?

Thank you for pointing this out. We will improve the SWE section by studying in more detail what factors may explain the classification. We will add information on forest cover fraction and topography to the analysis and investigate whether they have an impact on the classification. Also, we will add the ERA5-Land SWE dataset to the analysis as suggested in the previous comment.

**References**

D'Eon, R. G. (2004): Snow depth as a function of canopy cover and other site attributes in a forested ungulate winter range in southeast British Columbia, BC Journal of Ecosystems and Management, 3, 136–144.

Ishizaka, Masaaki & Motoyoshi, Hiroki & Yamaguchi, Satoru & Nakai, Sento & Shiina, Toru & Muramoto, Ken-ichiro. (2016). Relationships between snowfall density and solid hydrometeors, based on measured size and fall speed, for snowpack modeling applications. The Cryosphere. 10. 2831-2845. 10.5194/tc-10-2831-2016.

Kouki, K., Anttila, K., Manninen, T., Luojus, K., Wang, L., and Riihelä, A. (2019): Intercomparison of snow melt onset date estimates from optical and microwave satellite instruments over the northern hemisphere for the period 1982–2015, Journal of Geophysical Research: Atmospheres, 124, 11 205–11 219.

Lauri, T. (2010): Wind drift of snowfall between the radar volume and ground, Master thesis.

McNay, R. S., Peterson, L. D., and Nyberg, J. B. (1988): The influence of forest stand characteristics on snow interception in the coastal forests of British Columbia, Canadian Journal of Forest Research, 18, 566–573

Miller, D. H. (1964): Interception processes during snowstorms, vol. 18, Pacific Southwest Forest and Range Experiment Station, Forest Service, US.

Pirazzini, R. (2004): Surface albedo measurements over Antarctic sites in summer, Journal of Geophysical Research: Atmospheres, 109.

Pirazzini, R., Vihma, T., Granskog, M. A., and Cheng, B. (2006): Surface albedo measurements over sea ice in the Baltic Sea during the spring snowmelt period, Annals of Glaciology, 44, 7–14.

Vázquez-Martín, Sandra & Kuhn, Thomas & Eliasson, Salomon. (2021). Shape dependence of snow crystal fall speed. Atmospheric Chemistry and Physics. 21. 7545-7565. 10.5194/acp-21-7545-2021.

---

## Author Comment (AC2)

**Answers to referee #2**

We thank the reviewer for the encouraging review of our manuscript. Answer to the referee's comment is below. Our answer is marked with blue color.

The authors undertake a proof-of-concept investigation into how accurately optical satellite observations, namely Sentinel-2 surface reflectance-based grain size and microwave-based snow water equivalent (SWE) estimates can detect snowfalls over the Arctic. The technique developed by the authors is capable to detect at least 77% cases of snowfall using optical measurements alone in a correct way, which is a success taking into account that the method can be further elaborated (especially selecting a better cloud detection scheme (thermal infrared measurements?)). I would advice to retrieve the value of L from Eq. (4) and use it in the analysis. This is related to the fact that the conversion of L to SSA may lead to additional errors, which are difficult to assess. On the other hand, this shortcoming may not influence the results of this paper aimed not to the SSA determination but to the detection of snowfalls. The authors may comment on this issue in the paper. I advice the publication of this paper.

As it happens, we already retrieve the value of L from the Eq. (4). We will add a clarification about that to the text.